# A Data-Driven Model Predictive Control for Quadruped Robot Steering on Slippery Surfaces

**Paolo Arena** [1,*] , **Luca Patanè** [2,*] **and Salvatore Taffara** [1]

1 DIEEI, University of Catania, 95125 Catania, Italy; salvatore.taffara@phd.unict.it
2 Department of Engineering, University of Messina, 98122 Messina, Italy
* Correspondence: paolo.arena@unict.it (P.A.); lpatane@unime.it (L.P.)

**Abstract:** In this paper, the locomotion and steering control of a simulated Mini Cheetah quadruped robot was investigated in the presence of terrain characterised by low friction. Low-level locomotion and steering control were implemented via a central pattern generator approach, whereas high-level steering control manoeuvres were implemented by comparing a neural network and a linear model predictive controller in a dynamic simulation environment. A data-driven approach was adopted to identify the robot model using both a linear transfer function and a shallow artificial neural network. The results demonstrate that, whereas the linear approach showed good performance in high-friction terrain, in the presence of slippery conditions, the application of a neural network predictive controller improved trajectory accuracy and preserved robot safety with different steering manoeuvres. A comparative analysis was carried out using several performance indices.

**Keywords:** LMPC; NNMPC; CPG; quadruped robot; neural network; steering control





## 1. Introduction

This work is motivated by the necessity to explore the potentialities and limits of locomotion controllers typically used in the best-performing quadrupeds currently available and engaged to explore unstructured terrain. For these purposes, legged robots outperform wheeled machines, offering minimal evasiveness and superior climbing capabilities. This comes at an additional cost in terms of manoeuvrability (steering) and attitude control, in particular in the presence of slippery surfaces [1–3]. Gait parameters such as the step length and walking speed need to be carefully chosen. In the proposed work, a quadrupedal structure modelling the MIT Mini Cheetah robot was considered; the huge recent advances in its control architecture and its low cost make this structure an ideal candidate for the exploration of outdoor environments [4]. In the literature, the traditional ways to cope with the problem of slip detection and compensation are force-based and kinematic-based. The former [5,6] requires legs endowed with expensive and accurate force sensors, which can be damaged due to the repetitive impulse-like bumps in the touchdown phase. The latter [7] estimates slip and friction parameters through relative motion detection of the stance feet in the velocity space. In general, the stability of legged robots is dramatically compromised when walking on slippery surfaces due to the inertial effects caused by the sudden acceleration of the center of mass during leg touchdown events.

Legged machines have additional complexity due to the increased number of degrees of freedom to be controlled for locomotion efficiency trajectory tracking. With this aim, two main parallel controllers are implemented: one devoted to (high-level) trajectory following and the other related to the maintenance of a specified locomotion gait, i.e., a prescribed phase relation among the legs. One of the most used approaches for high-level control in legged machines relies on model predictive control (MPC), whereas the mostly used low-level locomotion control relies on the central pattern generator (CPG) approach. MPC is a well-known control strategy that has been studied deeply in recent decades

both in advanced research and in more industry-related applications[8], with interesting applications for online motion control of running bipeds on uneven terrain [9] or for gait adaptation in quadrupeds [10]. In particular, linear model predictive controllers (LMPCs) assume that the system dynamics can be efficiently approximated as linear. For certain applications involving highly nonlinear system control, nonlinear model predictive control (NMPC) is required [10–12], which comes at the expense of higher complexity in terms of design and implementation of the control law and stability analysis of the obtained closed-loop system.

Within the NMPC family, a neural network model predictive controller (NNMPC) is adopted when the modelling procedure is data-driven, and the system nonlinearities are assumed to be unknown [13,14]. Using this strategy, it is possible to build a black-box model of the system wherein the nonlinear characteristics can be learned from simulation and/or experimental data. Contrary to the standard approach, for which there is a need to estimate the friction coefficient, to cope with slippery surfaces, in this work, a neural model is shown to autonomously capture the nonlinear interactions between the ground reaction force under slippery conditions and the center of mass motion. This model can be used within the MPC controller to achieve efficient control of the robot motion.

Recently, high-level steering control was successfully addressed using an LMPC approach, assuming that the particular robot configuration would heavily simplify the robot dynamics (i.e., the concentration of all twelve actuators on the robot main body and the light leg inertia) [15].

However, this condition holds in the case of locomotion in terrain far from slippery conditions. Here, we show the limits of LMPC. Due to the complex inertia interplay, while moving on slippery surfaces, an accurate nonlinear analytical model of the real robot is particularly difficult to formalize, and black-box modelling strategies such as neural networks can be efficiently applied. On the other side, direct deployment of such control strategies in legged robots, especially when requested to move on slippery terrain, is very dangerous. As a consequence, the use of accurate dynamic simulation environments is essential for proper evaluation of robot performance and application limits.

The low-level CPG locomotion controller adopted in this work to generate a set of reference phase relationships among the actuators of the robot legs is based on the reaction-diffusion cellular neural network (RD-CNN) [16] paradigm, which can show steady-state stable, phase-shifted dynamics to generate the desired locomotion gait [17,18]. Further solutions based on embodied motor neurons have also been investigated [19]. Steering control is simply realised by applying suitable gains to the neuron links of the RD-CNN.

In previous work, the effect of linear and nonlinear MPC on quadrupedal steering was presented by the authors [20]. The adopted quadruped robot was controlled through a different and more complex CPG architecture based on the Matsuoka neuron model [21], the oscillations of which were synchronized using sensory feedback acquired using load sensors. MPC was adopted to provide a feedback signal to properly train the robot steering to follow a given path. In that study, the terrain was considered flat, and slippery conditions were not taken into account, which are the main focus of the proposed work.

Starting from the motivations mentioned above, the novelties of the proposed approach are summarized as follows:

- MPC is used to choose the steering command in order to maximise control performance. A careful comparison between LMPC and NNMPC is performed, demonstrating the limits of the former in the specific case of slippery terrain conditions. Here, the need to adopt a nonlinear MPC emerges in order to compensate for the complex dynamical effects involved in locomotion;
- The approach is data-driven and, as such, does not require an analytical model, which, in cases of complex multibody structures, is difficult to accurately derive. Moreover, the approach can be extended to other robot structures and scenarios with the same efficiency;

- Specific performance indicators are adopted during the different evaluation steps, including robot model selection (i.e., Akaike information criterion), steering controller evaluation (i.e., goodness of fit and slipping index), and evaluation of locomotion efficiency (i.e., stability and harmony [22,23]).

This paper is structured as follows. Section 2 shows the materials and methods considered, including the robotic neural locomotion architecture, the theoretical results on the stability of the steering locomotion gait, the MPC strategy, and the adopted performance indices. In Section 3 comparisons of LMPC and NNMPC implementations for different slippery scenarios are presented. Finally, conclusions are drawn in Section 4.

## 2. Materials and Methods

### 2.1. Quadrupedal Locomotion

Locomotion control in legged robots can be achieved through a CPG-based network devoted to generating rhythmic stereotyped patterns of activity [24]. In our Mini Cheetah quadruped robot, the CPG architecture was formalized by taking inspiration from previous structures developed for hexapod robots. According to [17], the neural network is arranged in the form of a tree-graph structure containing a number of neurons equal to the number of legs. Here, each neuron shows a steady-state nonlinear oscillation and is in charge of controlling the position of the corresponding leg joints through its state variables (Figure 1).

The dynamics of the single neuron for this particular application are described by the two following differential equations:

$$
\begin{cases}
\dot{x}_{1,i} = f_1(x_{1,i}, x_{2,i}) = -x_{1,i} + (1 + \mu)y_{1,i} - sy_{2,i} \\
\dot{x}_{2,i} = f_2(x_{1,i}, x_{2,i}) = -x_{2,i} + (1 + \mu)y_{2,i} + sy_{1,i} \\
y_{k,i} = \tanh(x_{k,i}), \quad k = 1, 2
\end{cases}
\tag{1}
$$

where parameters $\mu = 0.7$ and $s = 1$ were selected in order to obtain a stable limit cycle [17].

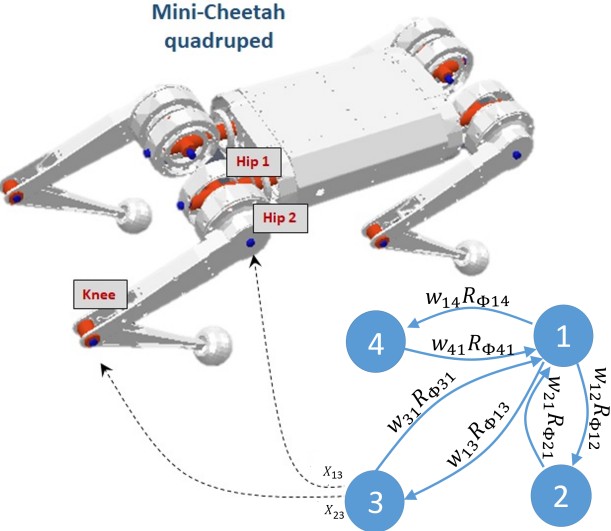

**Figure 1.** The Neural structure adopted for locomotion control. Each neuron is devoted to controlling the position of two leg joints (hip 2 and knee) through the two state variables, $x_{1,i}$ and $x_{2,i}$, defined in Equation (1).

The dynamics of the whole system can be therefore be represented in the following general form:

$$
\dot{\mathbf{x}}_\mathbf{i} = f(\mathbf{x}_\mathbf{i}) + k_L \sum_{i \neq j \, j \in N_r} [R_{ij}(w_{ij}\mathbf{x}_\mathbf{j}) - \mathbf{x}_\mathbf{i}]
\tag{2}
$$

where $f(\mathbf{x}_\mathbf{i}) = f(x_{1,i}, x_{2,i})$ represents the dynamics of the $i$-th uncoupled neuron, and $k_L$ is the feedback gain, whereas the summation operator represents the contribution of the

immediately neighbouring cells around cell $i$ within a neighbourhood ($N_r$). From the control point of view, this part represents the feedback error between the state variables of the $i$-th cell and the corresponding state variables of cell $j$, eventually after scaling through the weight ($w_{ij}$) and a phase shift with respect to cell $i$ via the rotation matrix $R_{ij}$.

The locomotion control strategy proposed in [17] allows for the definition of any type of locomotion gait through a specific array of phase shifts to be imposed on the robot legs.

$$\Theta_{gait} = [\theta_{FL,FL}; \theta_{FL,FR}; \theta_{FL,BL}; \theta_{FL,BR}]$$

in our application, and the trot gait was imposed via the following phase vector:

$$\Theta_{trot} = [0°; 180°; 180°; 0°] \tag{3}$$

As reported in [25], robot steering when following a path is performed by exploiting the $w_{ij}$ entries in Equation (1). In particular, only one gain value is calculated by the controller and applied to a subset of leg joints. In detail, in our case, $W = \{w_{i,j}\} \in \mathbb{R}^4$, where $i = \{1, 2\}$ and $j = \{FL, FR, BL, BR\}$. If the same scaling value ($\bar{w}_{ij}$) is imposed on the state variables of ipsilateral (e.g., right) legs, (right) steering is performed, maintaining the same gait (i.e., trot) [25]. In Figure 2, the signals related to the state variables acting on the hip 2 joints considering the four legs are shown when right steering is performed. As it is possible to notice, by modulating the weight ($w_{ij}$) entries related to ipsilateral legs, the state variable signal changes in amplitude, producing phase-invariant leg trajectory scaling.

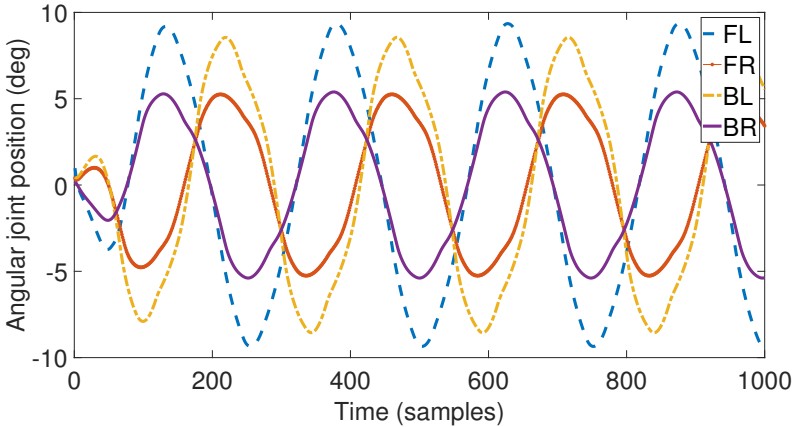

**Figure 2.** An example of weighted phase-shifted signals controlling robot steering via the $w_{ij}$ entries in Equation (2). In the legend FL, FR, BL, and BR correspond to the normalised first state variables of neurons 1, 2, 3, and 4, respectively, in Figure 1.

### 2.2. MPC Design

A standard MPC controller design phase was followed, both in the linear and nonlinear case. MPC makes use of a model of the system to be controlled, whereas in the linear case, it is feasible to derive an analytical model. In the nonlinear MPC approach, the model was identified using a data-driven approach, as often found in the literature. In our case, this is possible thanks to the availability of an accurate dynamic simulation environment, which allows for an implicit formulation of the robot model directly from data. This analytical model-free approach is therefore amenable to direct generalizations to other cases of study. In Figure 3 a high-level scheme of the system architecture developed in Matlab-Simulink is reported in the case of the NNMPC implementation. The only difference from the LMPC is the block outlined as "NN predictive controller", which, in the linear case, employs a linear model. The reference signal (*sig*) representing the desired yaw speed of the robot center of mass ($\omega_z$) is an input for the NN predictive control block, together with its actual value. The controller output consists of the gains ($w_{ij}$), which, applied in Equation (2), realize low-level steering control in the trotting Mini Cheetah. In our case, steering consists of scaling the state variables of the ipsilateral legs of an equal gain value; the controller output,

therefore, consists of a scalar value ($w$) applied equally to the joint state variables ($x_{ij}$) of ipsilateral legs. Communication with the CoppeliaSim dynamic framework was achieved using a 2-level s function available in Matlab.

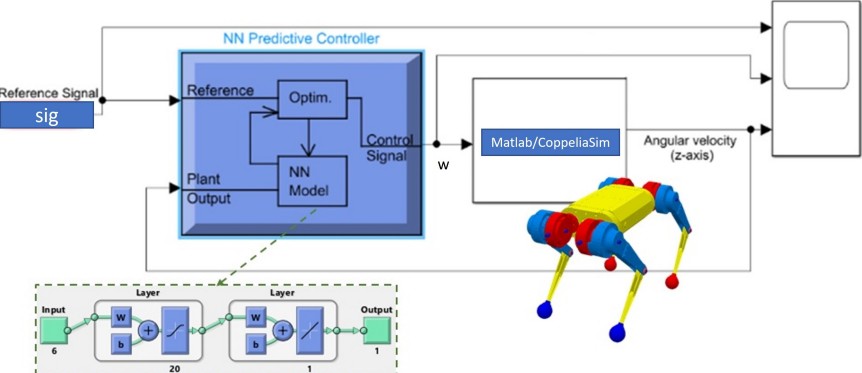

**Figure 3.** High-level scheme of the NNMPC-based architecture for steering control.

Within the NNPC block (Figure 3), the robot model receives the steering control signal as input (i.e., the gain ($w$) to be distributed among ipsilateral legs), whereas the output is the yaw velocity ($\omega_z$) of the robot center of mass. Once the model is obtained, the best control sequence is calculated by minimizing the following performance index ($J$) over the given horizon:

$$J = \sum_{j=1}^{N_H}(y_r(t+j) - y_m(t+j))^2 + \rho \sum_{j=1}^{N_U}(w'(t+j-1) - w'(t+j-2))^2 \qquad (4)$$

$$s.t. \ |w'(t)| < \gamma$$

where, according to the classical MPC formulation, $N_H$ is the prediction horizon where the tracking error is evaluated, $N_U$ is the control horizon where the control signal samples are analyzed, $w'$ is the tentative control signal, $y_r$ is the reference response, $y_m$ is the model output, $\rho$ (in this case, $\rho = 0.05$) weights the contribution of the sum of the squared control increments on the index ($J$), and $\gamma = 0.07$ is the saturation limit of the control input. The optimization block determines that control input ($w_0'$) minimizing $J$ is to be provided to the robot CPG controller as a steering command. The minimization algorithm used is *csrchbac*, a one-dimensional minimisation routine based on a backtracking technique [26]. According to the MPC guidelines, $N_H$ was chosen to have 20–30 samples covering the open-loop transient system response, while $N_U$ is a fraction (10–30%) of $N_H$. Typically, only the control input at the next time step is applied to the robot to make the control law more effective. The adopted MPC parameters are reported in Table 1.

**Table 1.** Parameters used in the NN predictive controller block are shown in Figure 3.

| Parameter | Value |
|---|---|
| Sample time | 0.05s |
| No. of manipulated variables | 1 |
| No. of measured outputs | 1 ($\omega_z$) |
| Prediction horizon ($N_H$) | 150 samples |
| Control horizon ($N_U$) | 40 samples |

Once the dataset is generated, the relation between steering command and angular velocity can be modelled using either a transfer function (LMPC) or a neural network (NNMPC).

*2.3. Performance Indicators*

Three classes of criteria were adopted to evaluate the control performance. The Akaike information criterion (AIC) was used to select the best model architecture in the linear and nonlinear cases. The selected two best models were compared using MSE and fit as performance indices.

On the other side, locomotion performance was assessed via the so-called "stability" and "harmony" indicators. In particular, a slipping index (SI) was used to quantify the locomotion reliability under slippery conditions. These quantities are introduced below.

2.3.1. Model Selection and Evaluation

Given a collection of data-driven models, the AIC was adopted [27,28] to estimate prediction quality based on prediction accuracy and model complexity. The best model has the smallest AIC index, defined as follows:

$$AIC = N \cdot \log \left( \det \left( \frac{1}{N} \sum_{1}^{N} \epsilon(t, \hat{\theta}_N) \left( \epsilon(t, \hat{\theta}_N) \right)^T \right) \right) + 2n_P + N \cdot \left( n_y \cdot (\log(2\pi) + 1) \right) \quad (5)$$

where $N$ is the number of samples in the estimation dataset, $\epsilon(t)$ is a $n_y - by - 1$ vector of prediction errors, $\theta_N$ represents the estimated parameters, $n_p$ is the number of parameters to be estimated (i.e., model complexity), and $n_y$ is the number of model outputs. The AIC is currently used as a performance index suited for the evaluation of real-time processes. It considers both the prediction accuracy and the model complexity; therefore, minimizing the AIC index guarantees a good compromise between performance and computational resources to be dedicated to accomplishing the steering control task presented herein.

After selecting the best linear and nonlinear model from the AIC analysis, the two models were compared by computing the mean squared error (*MSE*) and the goodness of fit (*Fit*) index. The fit index represents the error norm between the model output (*x*) in the testing dataset and the corresponding reference signal ($x_{ref}$). It is based on the normalised root mean square error (NRMSE) [29,30]:

$$Fit = \left( 1 - \frac{||(x_{ref} - x)||}{||(x_{ref} - \bar{x}_{ref})||} \right) \cdot 100 \quad (6)$$

2.3.2. Locomotion Performance Evaluation

For locomotion performance evaluation, two different performance indices were used: stability and harmony [22,23]. Stability accounts for robot deceleration and rotational oscillation. It is defined as follows:

$$Acc_{min} = |\min(Acc)|$$
$$Ang_{PP} = \max(Ang) - \min(Ang)$$
$$a_{min} = \max(Acc_{min})$$
$$PP_{min} = \max(Ang_{PP})$$
$$A_{min_{final}} = \max(A_{min})$$
$$PP_{min_{final}} = \max(PP)$$
$$Stability = \frac{e^{-\lambda A_{min_{final}}} + e^{-\lambda PP_{min_{final}}}}{2} \quad (7)$$

where *Acc* indicates the accelerations of the robot body, and *Ang* indicates the angular velocities in all three axes. The values of the *Acc* minimum peak ($Acc_{min}$) and *Ang* from peak to peak ($Ang_{PP}$) characterize the locomotion stability. Stability is evaluated during the time window of the experiment, which is divided into several cycles or periods ($N_{periods}$).

We define $A_{min} = \{a_{min,1}; a_{min,2}; \dots; a_{min,N_{periods}}\}$, $PP_{min} = \{PP_{min,1}; PP_{min,2}; \dots; PP_{min,N_{periods}}\}$. The decay constant ($\lambda$) is fixed to $\lambda = 0.1$.

Harmony indicates the adaptation of the robot body position to the change in the terrain level. To evaluate the harmony index, the following relations are calculated:

$$
\begin{aligned}
RI_{a_{min}} &= \frac{\min(Acc_{min})}{\max(Acc_{min})} \\
RI_{PP_{min}} &= \frac{\min(Ang_{PP})}{\max(Ang_{PP})} \\
RI_{A_{final}} &= \min(RI_A) \\
RI_{PP_{final}} &= \min(RI_{PP}) \\
Harmony &= \frac{RI_{A_{final}} + RI_{PP_{final}}}{2}
\end{aligned}
\tag{8}
$$

where $RI_A = \{RI_{a_{min,1}}; RI_{a_{min,2}}; \ldots; RI_{a_{min,N_{periods}}}\}$ and $RI_{PP} = \{RI_{PP_{min,1}}; RI_{PP_{min,2}}; \ldots; RI_{PP_{min,N_{periods}}}\}$.

The optimal condition consists of high values for both stability and harmony (with unity upper bound). It should be noted that the intrinsic nature of legged locomotion introduces angular oscillations in the robot yaw, and the presence of an active steering controller is fundamental when forward path following is requested.

In fact, the typical cyclic actuation of the legs and the alternation of stance and swing phases unavoidably introduce periodic pulses, especially during the touchdown event, which, filtered by the dynamics structure, lead to oscillations in the motion of the robot centre of mass. Therefore, even while moving on perfectly flat terrain, the presence of angular oscillations due to this endogenous dynamic is evident. Typically, this is considered a natural effect of legged locomotion that needs to be taken into account and maintained below a certain threshold.

### 2.4. Performance under Slippery Conditions

Slippery conditions occur whenever a foot in the stance phase shows a non-zero velocity with respect to the world reference frame [7]. To quantify the efficiency of the controller under slippery conditions with respect to different friction conditions, the *slipping index* (*SI*) is defined as follows:

$$
SI = \|P_{fs}^W(t) - P_{fs}^W(t-1)\|
\tag{9}
$$

where $P_{fs}^W(t)$ is the planar vector connecting the generic robot foot position when in the stance phase to the world reference frame (*W*). The index takes into account the difference between two consecutive planar positions of a specific foot in contact with the ground. Considering the time between two consecutive position recordings as unitary, *SI* accounts for the residual speed of the robot foot. The idea behind this definition is that the generic foot, while in a stance, should not move under ideal conditions, even in cases of high friction. In this paper, since a simulation approach is adopted, this index can be reliably evaluated with respect to the world reference frame (*W*). In the case of a real robot, it can be easily referred to as the robot reference frame using well-known kinematic relations. Under real conditions, as well as when using a realistic dynamic simulation environment, the index never goes to zero but is maintained around a certain small quantity, recording small natural fluctuations. Therefore, if *SI* is maintained below a certain upper bound ($\overline{SI}$), the slipping conditions can be neglected; otherwise, slipping has to be considered a source of potential instability. Due to the dynamic conditions, this index has to be experimentally tuned both in simulations and in real robot deployment.

### 2.5. Simulation Environment

The software tools employed to perform the quadruped robot simulations are Matlab, Simulink, and CoppeliaSim. In particular, the control scheme used to drive the robot is

implemented in Simulink. The dataset acquisition campaigns, the model generation for the NMPC and LMPC, and the locomotion gait generation are implemented in Matlab. CoppeliaSim is a dynamic simulation environment in which the legged robot and its interaction with the adopted terrain can be modelled, simulated, and controlled through Matlab [31].

Figure 4 shows the steering behaviour of the Mini Cheetah model within the simulated environment in which the tests were performed. In Table 2, the physical characteristics of the modelled robot are reported. The adopted dynamic simulation environment allows for the integration of all the physical parameters of the robot, including the characteristics of the actuators equipped on the real counterpart, such as the PID parameters for low-level position control and maximum torque.

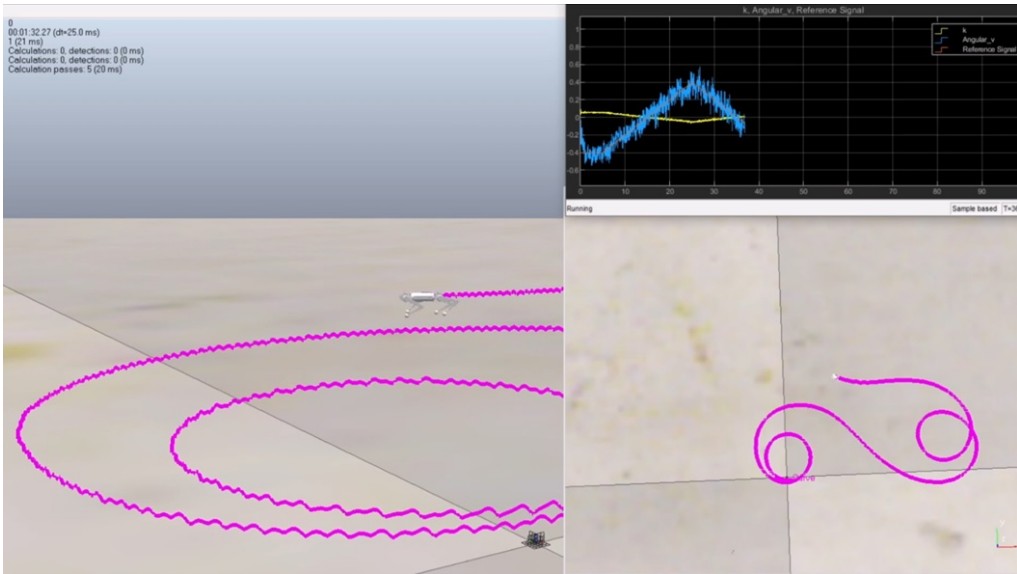

**Figure 4.** The simulated Mini Cheetah robot is shown while performing steering manoeuvres in the CoppeliaSim framework. A trace of the center-of-mass position is left by the robot during the simulation, as shown in the lateral (left panel) and top (bottom-right panel) views of the scene. The reference signal and the actual steering speed are also shown in the top-right panel.

**Table 2.** The physical characteristics of the Mini Cheetah robot adopted in the simulated model.

| Feature | Value |
| --- | --- |
| Height | 30 cm |
| Length | 48 cm |
| Width | 27 cm |
| Weight | 9 kg |

The effect of gain-based steering control in terms of steering velocity is reported in Figure 5; a portion of the learning dataset (composed of a total of 500 steps with random amplitude and duration) is shown under normal friction conditions that correspond to the interaction between rubber and dry asphalt.

Different behaviour is observed in the presence of low friction that corresponds to the interaction between rubber and wet ice, as shown in Figure 5b. The complexity of the low-friction model hidden within the data is evident when linear and nonlinear modelling techniques used in the design of the MPC are compared.

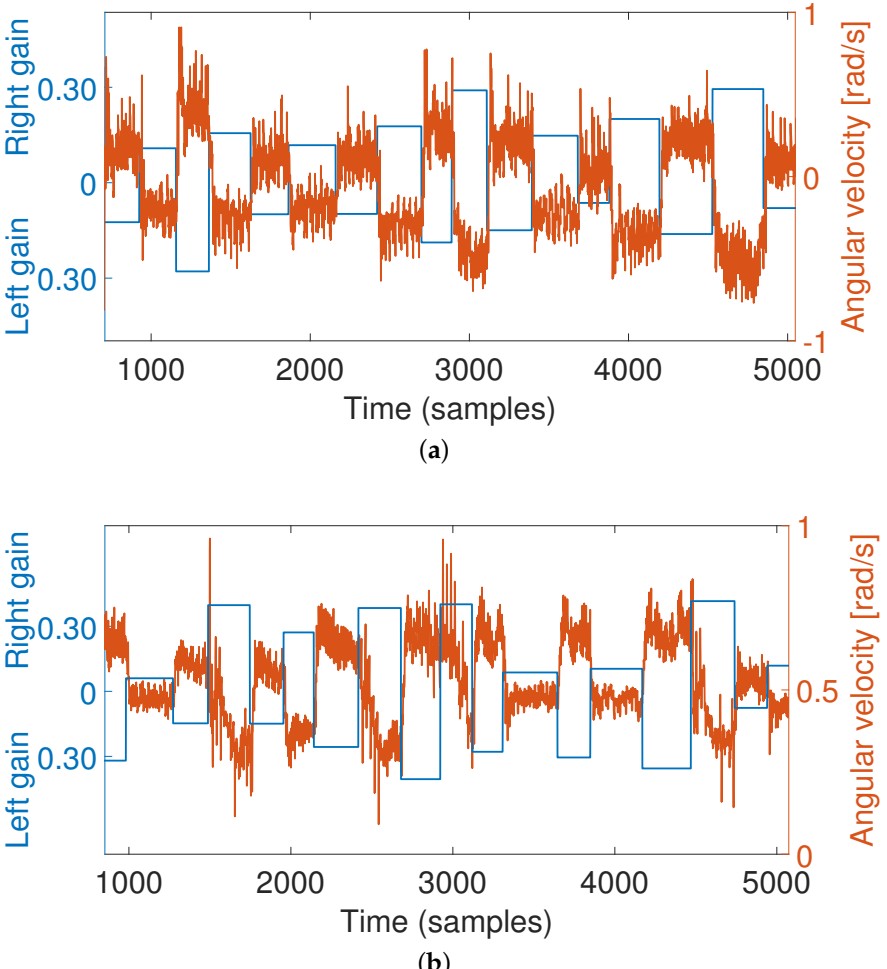

**Figure 5.** Dataset obtained in the dynamic simulation environment in which couples ($w$-$\omega_z$) are collected from the walking quadruped robot: (**a**) normal friction; (**b**) low friction.

## 3. Results

### 3.1. Data-Driven Robot Model

The development of the robot model for MPC-based steering control was carried out using a set of transfer functions with different numbers of poles and zeros for the linear case, whereas in the nonlinear case, different I/O regressor couples were analyzed as reported in Table 3. The number of hidden neurons in the neural network was fixed to 20 through a searching strategy based on a combination of expert knowledge to fix a rough searching domain and a grid search to identify the best configuration in terms of prediction accuracy on the validation dataset.

The AIC analysis for all the models in Table 3 is shown in Figure 6. Therefore, the optimal model structures selected are 5-1 (normal friction) and 3-2 (low friction) in the linear case, whereas, in the neural network modelling, the selected configurations are 2-4 (normal friction) and 4-5 (low friction).

In Table 4, the linear and nonlinear model *Fit* values and the Pearson correlation coefficient ($R$) are reported using the optimal structures indicated by the AIC analysis.

According to the analysis of the Fit values, it is evident, as expected, that the nonlinear approach outperforms the linear approach. Furthermore, applying a neural network to model the robot behaviour allows comparable performance to be obtained in the scenarios with both standard and low friction, whereas the linear solution shows a significant decrease in accuracy when the low-friction scenario is considered.

**Table 3.** Transfer function and I/O regressors for the linear and nonlinear case, respectively. The optimal structure was selected through the AIC index.

| | LMPC | | NNMPC |
| --- | --- | --- | --- |
| № | TF Poles-Zeros | № | I/O Regressors |
| 1 | 2-1 | 1 | 1-3 |
| 2 | 3-1 | 2 | 2-3 |
| 3 | 4-1 | 3 | 2-4 |
| 4 | 5-1 | 4 | 3-4 |
| 5 | 3-2 | 5 | 3-5 |
| 6 | 4-2 | 6 | 4-5 |
| 7 | 5-2 | 7 | 4-6 |
| 8 | 4-3 | 8 | 5-6 |
| 9 | 5-3 | 9 | 5-7 |
| 10 | 5-4 | | |

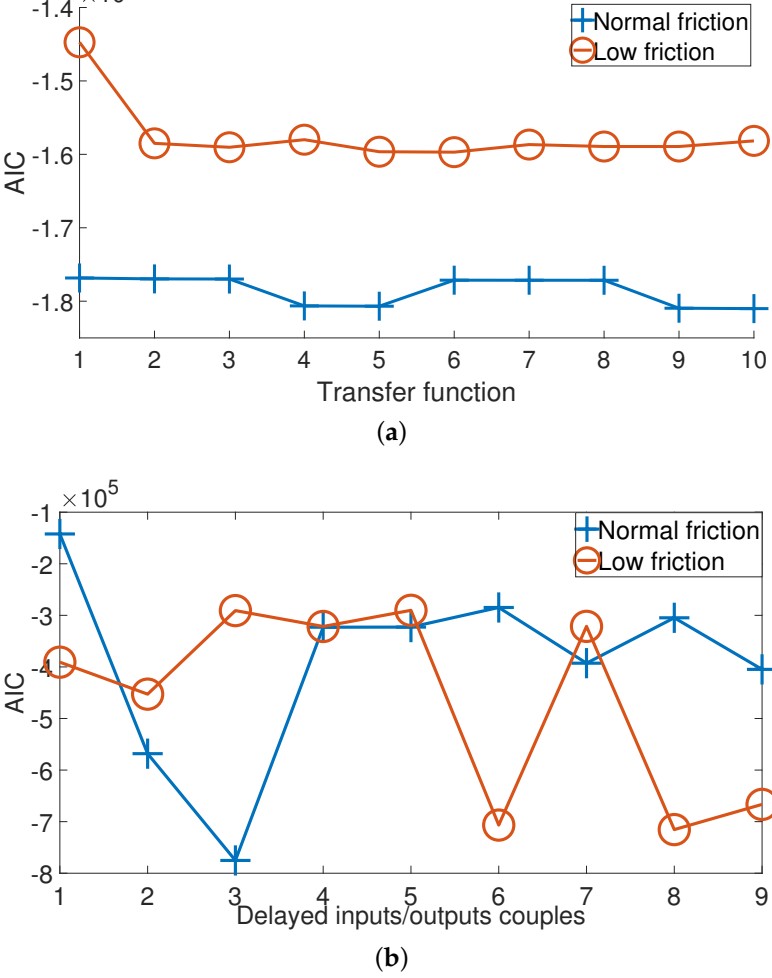

**Figure 6.** AIC index results: the linear (**a**) and nonlinear (**b**) architectures were considered in the presence of normal and low friction. The numbers reported on the x-axis correspond to the configurations described in Table 3.

Figure 7 shows the estimation accuracy of the linear and nonlinear models for different friction conditions between the robot and the terrain. It can be noticed that the angular velocity signals obtained at low friction are less regular than in the normal-friction case. However, the neural-network-based model can correctly estimate the output signals under both conditions.

**Table 4.** Correlation coefficients and *Fit* values between the model output and the actual value for the linear and nonlinear cases in the presence of normal and low friction.

| Model | Friction | Fit | R |
|---|---|---|---|
| Linear | normal | 61.67 | 0.91 |
| | low | 55.80 | 0.89 |
| Nonlinear | normal | 76.97 | 0.97 |
| | low | 75.10 | 0.97 |

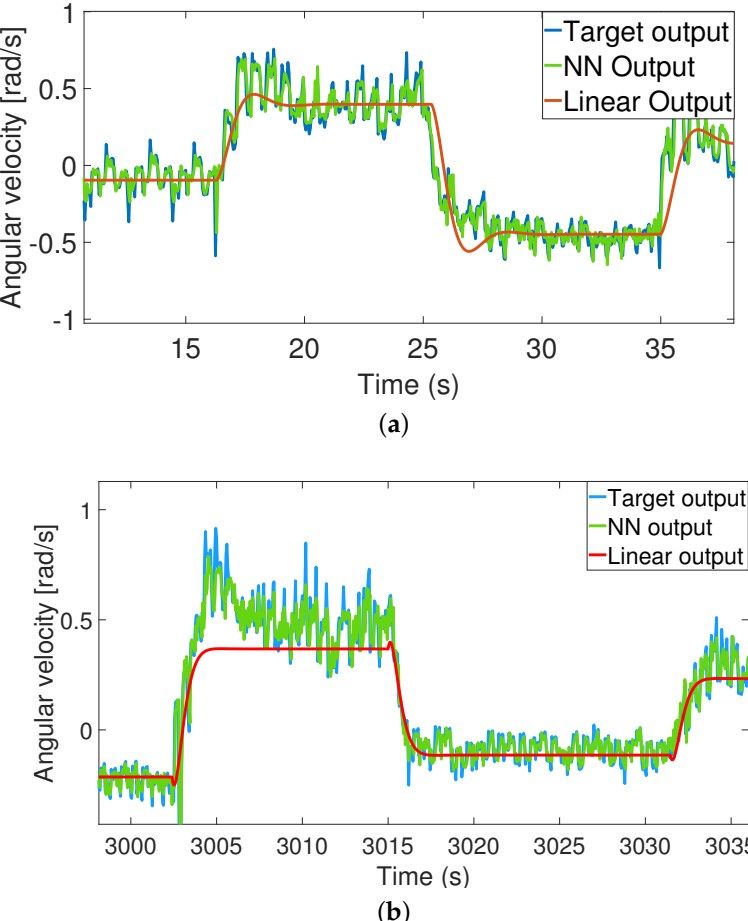

**Figure 7.** Comparison between the angular velocity acquired from the simulated robot while providing different steering commands (i.e., target output) and the behaviour of the learned models (i.e., linear and neural) subject to the same command signals. The scenarios with (**a**) normal and (**b**) low friction are presented.

A statistical analysis of the distributions of the error between the model output and the target signal is reported in Figure 8, where a typical Gaussian distribution is shown in both cases. The improved performance of the nonlinear model is demonstrated by the reduced variance of its distribution compared to the linear approach. In the case of low friction, the larger variance seems therefore to produce larger average errors, which contribute to boosting the nonlinear effects, which affect the overall robot behavior. Time comparison among the models (Figure 7) reveals that the linear model succeeds in acquiring the average linear dynamics hidden in the nonlinear system, whereas the nonlinear model succeeds in capturing more details of the angular velocity dynamics, which are typical of quadrupedal locomotion. These are instead treated as errors and filtered out by the linear model. These aspects are not detectable in the statistical plots shown in Figure 8. Moreover,

the linear model seems to follow the system with a certain delay that is clearly visible during the transient phases. This aspect can be negligible in the case of normal friction but can become critical in the case of low friction, where it is strictly needed to capture the high-frequency reactions of the robot. These modelling aspects become essential for the success of nonlinear control.

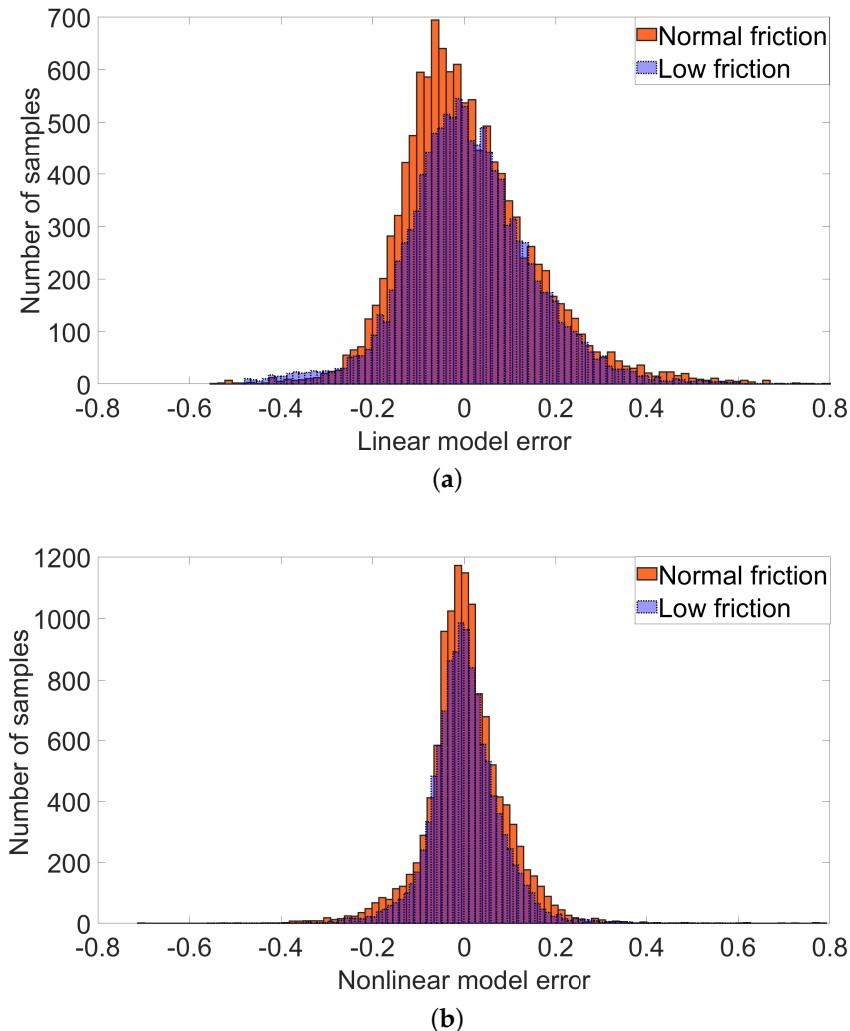

**Figure 8.** Statistical analysis: (**a**) error distribution for the (**a**) linear and (**b**) nonlinear models the in presence of normal and low friction.

*3.2. MPC-Based Steering Control*

Three different reference signals were considered to control the robot steering: a sine wave, a sequence of steps, and a triangle reference.

Figure 9 shows the results obtained from the dynamic simulation of the controlled quadruped robot for the different reference signals adopting the LMPC and the NNMPC in the scenarios with normal friction. The behaviour of the robot in the presence of a more challenging scenario with low friction is reported in Figure 10. The low-friction conditions lead to inertial disturbances, which the LMPC is not able to compensate for, whereas the neural-based MPC succeeds in stabilizing. The high-frequency oscillations around the reference value that are typically present in legged locomotion are emphasized by the presence of low friction, although the NNMPC is demonstrated to guarantee robot stability in the reported experiments conducted using a realistic dynamic simulation environment.

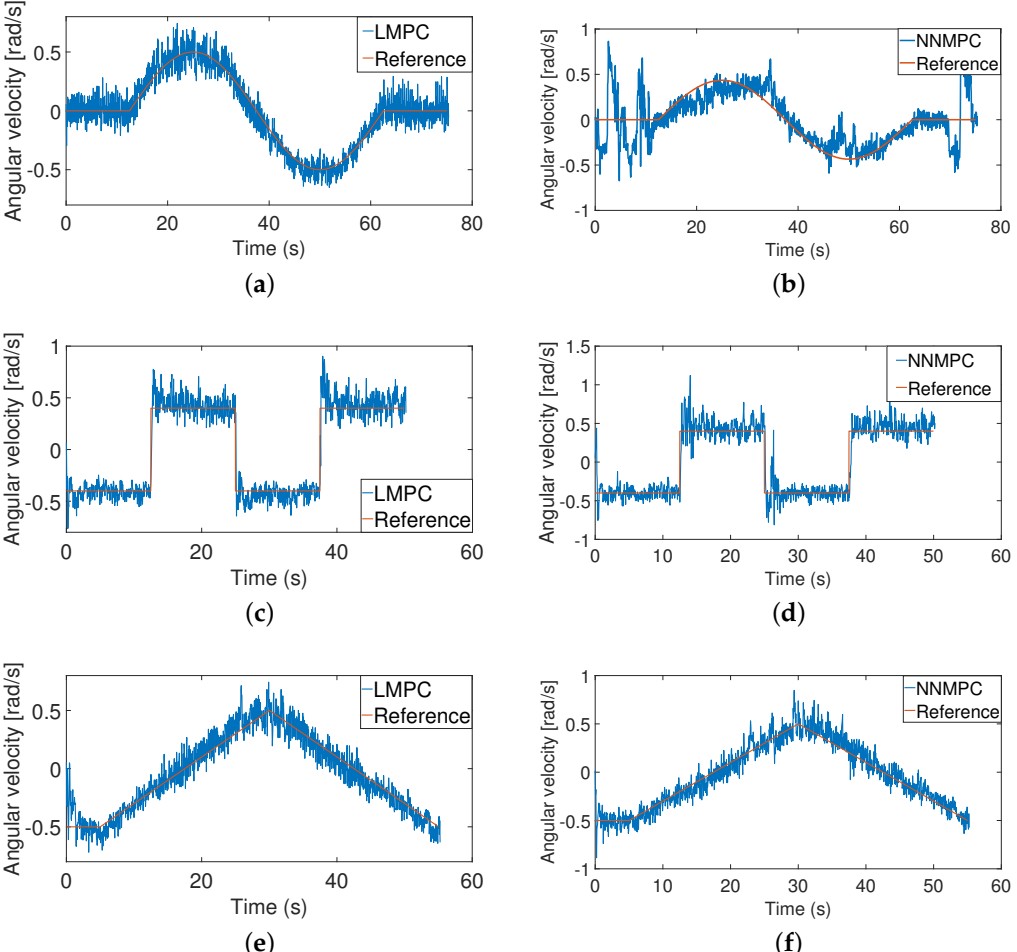

**Figure 9.** Results obtained using the LMPC and NNMPC under normal friction conditions: (**a**,**b**) sine wave reference; (**c**,**d**) step reference; (**e**,**f**) triangle reference.

As reported in Section 2.4, the slipping index was introduced as a simple method to statistically quantify the slippery effect, which can cause robot failure. As already discussed, its evaluation requires the tuning of a threshold ($\overline{SI}$). This can be obtained through a statistical analysis performed on the *SI* distribution, as depicted in Figure 11. Here, *SI* is reported for the FR leg, while in the stance phase, it is recorded during walking. Figure 11a,b reports the values obtained using the LMPC and the NNMPC when a zero reference signal (i.e., forward path) is provided to the system. When the robot follows a simple forward trajectory, both controllers achieve equivalent performance. In the case of more complex reference paths, such as a sine signal, the weakness of LMPC arises, as shown in Figure 11d. Here, the LMPC and NNMPC controllers are compared in the low-friction scenario, acquiring samples of *SI* soon before the falling event shown in the LMPC case (Figure 10a). In Figure 11c, the *SI* shows a large statistical difference and is much smaller in the NNMPC application. In particular, as a result of this statistical analysis, a safe threshold to be used is $\overline{SI} = 0.02$, which is quantified as the boundary of the third quartile around the median *SI* value in the LMPC application. This can be considered a maximum normalised speed that, if overcome, can cause the robot to fall.

In the presence of a typical robot ground interaction with normal friction, the performance of the NNMPC is mostly comparable with linear MPC, as reported in Table 5. A normal friction parameter helps to maintain the robot's stability while steering, reducing the need for a nonlinear approach.

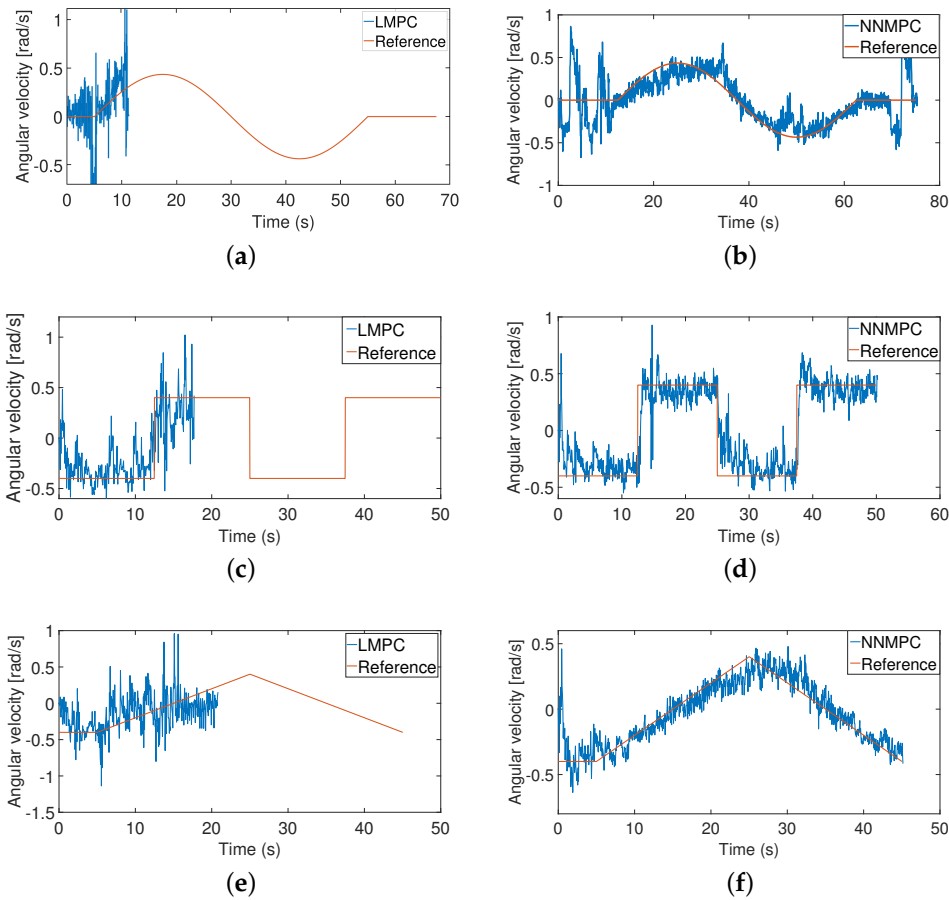

**Figure 10.** Results obtained using the LMPC and NNMPC under low-friction conditions: (**a**,**b**) sine wave reference; (**c**,**d**) step reference; (**e**,**f**) triangle reference.

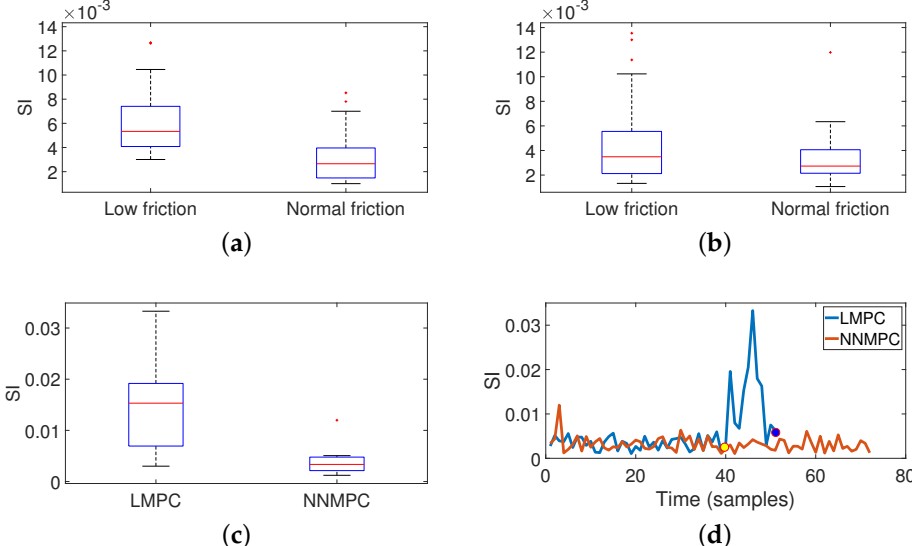

**Figure 11.** Slipping statistical analysis: (**a**,**b**) LMPC and NNMPC cases, respectively, when the robot follows a zero reference signal; (**c**) SI distribution when the robot follows a sine reference (Figure 10a,b) in the low-friction scenario. (**d**) SI time evolution related to the previous case. The yellow circle represents the moment at which the robot starts to oscillate, and the blue circle represents the moment when the robot has fallen. Within each box, the central mark is the median, the edges are the 25th and the 75th percentile, the whiskers extend to the most extreme data points that the algorithm considers not to be outliers, and the outliers are depicted individually as '+'.

**Table 5.** *Fit* and *MSE* values obtained using the LMPC and NNMPC approaches with three different reference signals.

| Ref. Signal | Fit | | MSE | |
|:---:|:---:|:---:|:---:|:---:|
| | **LMPC** | **NNMPC** | **LMPC** | **NNMPC** |
| Triangle | 70.75 | 70.97 | 0.013 | 0.009 |
| Sine wave | 68.40 | 67.45 | 0.008 | 0.01 |
| Steps | 63.99 | 62.69 | 0.009 | 0.02 |

The performance of the LMPC drastically degrades in the presence of scenarios with low friction in the foot–ground interaction. As shown in Figure 10, in this case, when the linear model is adopted, the robot is no longer able to correctly follow the reference steering signals, falling on the ground. The NNMPC is instead able to complete the trials, although the fitting between the reference and the current signal decreases compared with the normal friction case. The performance indices for the low-friction scenario, when the NNMPC is adopted, are reported in Table 6.

**Table 6.** *Fit* and *MSE* values obtained using the NNMPC approach with three different reference signals.

| Reference Signal | Fit | MSE |
|:---:|:---:|:---:|
| Triangle | 59.12 | 0.02 |
| Sine wave | 56.79 | 0.04 |
| Steps | 55.59 | 0.03 |

In Figure 12, the stability and harmony indices are reported, showing the statistical distribution over 20 trials. Here, the robot follows a straight path considering the normal and low-friction environments and applying the LMPC and NNMPC control strategies.

The stability and harmony indices for both the LMPC and NNMPC strategies are significantly better in the normal friction scenario than on slippery terrain. The statistical relevance of the different distributions was analyzed in Table 7 using Welch's *t*-test, resulting in a test decision for the null hypothesis. Considering the results, *h* is 1 if the test rejects the null hypothesis at the 5% significance level and 0 otherwise. LMPC and NNMPC show the same distribution only for the stability analysis under normal friction conditions, whereas in all the other cases, the differences are statistically relevant and demonstrate the effectiveness of the NNMPC if compared with its linear counterpart.

**Table 7.** Welch's *t*-test was applied to the stability and harmony indices reported in Figure 12; the subscripts indicate normal friction (NF) and low friction (LF).

| | Stability | | Harmony | |
|:---|:---:|:---:|:---:|:---:|
| | *h* | *p-Value* | *h* | *p-Value* |
| **LMPC**$_{NF}$ **vs. LMPC**$_{LF}$ | 1 | 0.0001 | 1 | 0.000005 |
| **NNMPC**$_{NF}$ **vs. NNMPC**$_{LF}$ | 1 | 0.001 | 1 | 0.00006 |
| **LMPC**$_{NF}$ **vs. NNMPC**$_{NF}$ | 0 | 0.82 | 1 | 0.001 |
| **LMPC**$_{LF}$ **vs. NNMPC**$_{LF}$ | 1 | 0.01 | 1 | 0.04 |

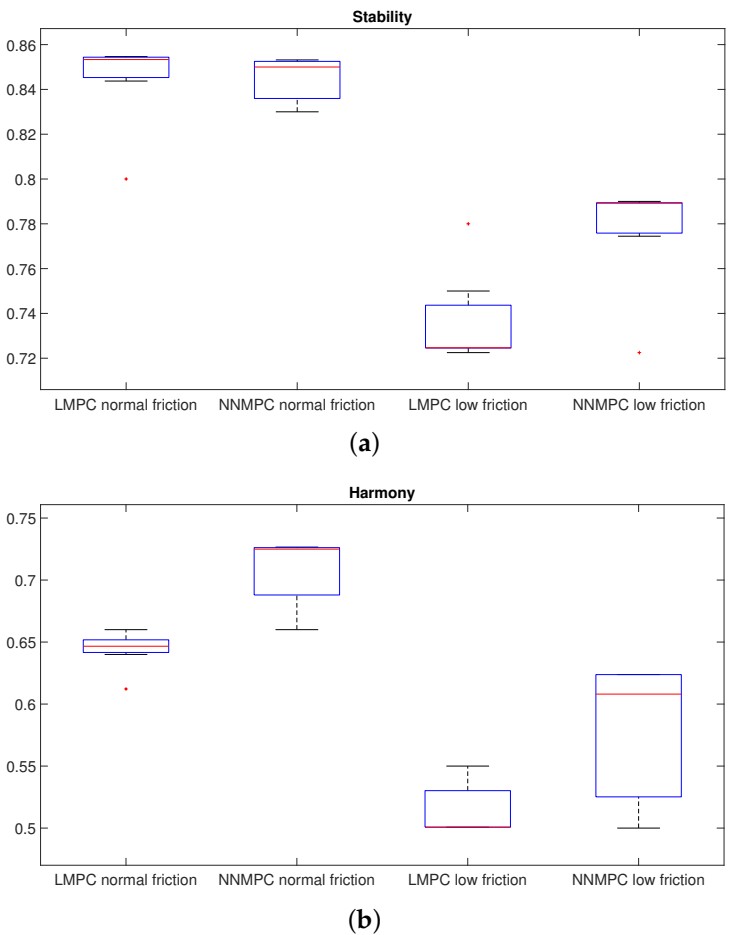

**Figure 12.** Performance indices when the robot follows a straight path considering the low-friction environment and applying the LMPC and NNMPC control strategies: (**a**) stability and (**b**) harmony. The statistics were determined over 20 trials, changing the robot initial leg configuration for each case.

## 4. Conclusions

In this paper, locomotion control in a simulated Mini-Cheetah quadruped robot moving in a slippery terrain was deeply analysed. The overall locomotion strategy was considered a hierarchical control task. The low-level phase-shifted synchronization among the robot legs, for general manoeuvres including steering, was entrusted to a CPG.

For high-level trajectory control, linear and nonlinear (neural network-based) MPCs were compared. MPC guides robot steering based on a reference consisting of the angular velocity (yaw speed) and acting on specific gains modulating the neural signals applied as position control references to the robot joints.

To properly compare the results, a neural network and a linear transfer function model were developed and optimized, using a data-driven approach to model the quadruped robot behaviour. The adopted dataset was generated using a model of the Mini Cheetah robot simulated in a dynamic environment, both under typical working conditions and in the presence of slippery terrain. The results obtained with NNMPC were compared with the linear MPC-based approach. The selection of the optimal model was achieved in both cases through the *AIC* index, taking into account both the accuracy and the complexity of the model. A comparative analysis was carried out taking into account the *Fit* and the *MSE* indices. The difference between the two control systems is evident in the case of relevant slippery conditions: the LMPC was no longer able to complete the requested steering trajectories, causing a robot fall. The main reason for this result is that the impulsive forces generated on the robot body during the leg touchdown and stance phase can be filtered out only in the case of a normal friction condition. In this case, a linear model can

sufficiently approximate the robot dynamic response to the steering commands. In the case of slippery surfaces, the advantage of an NNMPC is evident. The quality of the control architecture was also demonstrated using typical indices adopted for legged locomotion, such as stability and harmony. These quantified improvements obtained with NNMPC control in the low-friction case. The dynamic simulation analysis of the quadruped robot represents a required step for future implementations on the hardware prototype. Particular attention has been devoted to reducing the computational complexity in view of an onboard implementation for both the developed CPG and the MPC through the selection of optimal linear and nonlinear robot models. The adopted approach, which is essentially data-driven, can be extended to different scenarios, including uneven terrain and robot architectures, as discussed in the manuscript. Further research activities will be devoted to the integration of additional leg segments inspired by an insect tarsus, which improves the adhesion capabilities through different mechanical solutions, including the presence of claws.

**Author Contributions:** Conceptualization, methodology, software, investigation, and writing: L.P. and S.T.; Conceptualization, supervision, writing—review and editing: L.P. and P.A.; Supervision and project administration: L.P. and P.A. All authors have read and agreed to the published version of the manuscript.

**Funding:** The work was supported PNRR MUR project PE0000013-FAIR.

**Data Availability Statement:** The data presented in this study are available on request from the corresponding author.

**Conflicts of Interest:** The authors declare no conflict of interest.

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
