# Peer review of "A Data-Driven Model Predictive Control for Quadruped Robot Steering on Slippery Surfaces"

_robotics, doi:10.3390/robotics12030067_

Round 1

Reviewer 1 Report

In this paper, the locomotion and steering control of a simulated Mini Cheetah quadruped robot was investigated in presence of terrains characterized by low friction. In the simulation environment, the central pattern generator method is used to realize the low order motion and steering control, and the neural network and linear model predictive controller are used to realize the high order steering control. The results show that although the linear method performs well in high friction terrain, the application of the neural network predictive controller improves the trajectory accuracy in slippery conditions. This method is of great research significance, but I still have some questions:

1. What is the basis for the evaluation criteria of AIC construction in equation (5) in 2.3.1? Please provide a detailed explanation of the benefits of such construction.

2. In Figure 10, although NNMPC has a better effect than LMPC, there are still significant errors at some points in (d). Will this result affect the application of actual robots?

3. The verification methods of the whole paper are implemented in the simulation environment, and many results have large errors. Should different elements of actual robots be integrated into the simulated environment as much as possible, such as friction and transmission efficiency? These can better verify the practicability of the method.

4. There are some formatting problems in the paper. For example, lines 52 and 53 are not indented after wrapping.

Author Response

Reviewer #1: In this paper, the locomotion and steering control of a simulated Mini Cheetah quadruped robot was investigated in presence of terrains characterized by low friction. In the simulation environment, the central pattern generator method is used to realize the low order motion and steering control, and the neural network and linear model predictive controller are used to realize the high order steering control. The results show that although the linear method performs well in high friction terrain, the application of the neural network predictive controller improves the trajectory accuracy in slippery conditions. 

Authors: We thank the Referee for the time spent in reviewing our manuscript. We performed a careful revision of the paper to improve readability, better explaining the adopted techniques and the obtained results. Detailed answers to each aroused point are reported in the following.

Reviewer #1: This method is of great research significance, but I still have some questions:

  1. What is the basis for the evaluation criteria of AIC construction in equation (5) in 2.3.1? Please provide a detailed explanation of the benefits of such construction.

Authors: The AIC Akaike Information Criterion is currently used as a performance index suited for the evaluation of real time processes. It, on the one hand, considers the prediction accuracy, (i.e.) the first term of the index in Eq.(5), which accounts for the quadratic time averaged matrix of output residue. This has to be counterbalanced by a term (the following in the same equation) which accounts for the model complexity, i.e, the number of parameters involved in the model and the weighted number of model outputs. So, a minimum AIC index is a good compromise between accuracy and complexity.    

Reviewer #1:

  1. In Figure 10, although NNMPC has a better effect than LMPC, there are still significant errors at some points in (d). Will this result affect the application of actual robots?

Authors: The reviewer is right: especially in the time window between 12 and 25 sec, there are certain abrupt overshoots, but the benefits of the controller are here appreciated. In fact, the low friction conditions lead to inertial disturbances, which the LMPC is not able to compensate for, whereas, at least in all the simulations performed in our realistic simulation environment, the neural based MPC succeeds in stabilizing.  Considering the real implementation, the authors believe that the NN model should be properly refined considering real data coming from the prototype, obtaining a model tailored on the robot specific structure. In these conditions the degree of disturbance rejection would be efficient. 

Reviewer #1:

  1. The verification methods of the whole paper are implemented in the simulation environment, and many results have large errors. Should different elements of actual robots be integrated into the simulated environment as much as possible, such as friction and transmission efficiency? These can better verify the practicability of the method.

Authors: Our simulated model derives from importing all the essential robot components, including masses and inertial effects. In our simulation, the connections among the links take place through actuated joints which can be modelled through a  spring-mass-damper dynamics. These parameters represent the main mechanical aspects that take into account both the friction and the transmission efficiency and can be identified according to the real robot specifications.  Some sentences on this aspect were introduced in the revised manuscript

Reviewer #1:

  1. There are some formatting problems in the For example, lines 52 and 53 are not indented after wrapping.

Authors: Thanks. We revised the paper as suggested.

Reviewer 2 Report

The paper addresses an interesting topic in robotics, namely a data-driven predictive control model implemented on a quadruped robot.

The abstract is well done.

As far as keyword abbreviations are concerned, I note that they are known and used in articles, so I recommend leaving them as such. An MPC controller is designed.

As correction suggestions I have the following:

-on the left side of table 1 there is a dot. I think it should be removed!

- The AIC indices in fig. Is it normal to have that big gap between the values?

- how will this study improve field behaviour in frictionless conditions? what is that environment (boot?)

-if there are frictionless ground conditions, can't the underfoot be designed in such a way to improve contact?

Author Response

Authors’ Reply to Reviewer 2’s comments

Reviewer #2: The paper addresses an interesting topic in robotics, namely a data-driven predictive control model implemented on a quadruped robot. The abstract is well done. As far as keyword abbreviations are concerned, I note that they are known and used in articles, so I recommend leaving them as such. An MPC controller is designed.

Authors: We thank the Referee for the time spent in reviewing our manuscript. We performed a careful revision of the paper to improve readability, better explaining the adopted techniques and the obtained results. Detailed answers to each aroused point are reported in the following.

Reviewer #2: As correction suggestions I have the following:

- on the left side of table 1 there is a dot. I think it should be removed!

Authors: The Reviewer is right, we fixed the problem.

Reviewer #2: The AIC indices in fig. Is it normal to have that big gap between the values?

Authors: Fig. 6 is a proof that, as far as the LMPC is used, the performance in low friction are always worst than in the NMPC case, for any system order. This is properly testified from the gap. On the other side, the NMPC (Fig.6b) does not reveal such a problem, and the worst NNMPC case is better than the best LMPC AIC index.    This issue was clarified in the revised version of the manuscript.

Reviewer #2: how will this study improve field behaviour in frictionless conditions? what is that environment (boot?)

Authors: The authors performed comparison in cases of high and low friction. The former was representative of the contact among rubber and concrete whereas the latter of a contact among rubber and wet ice. We demonstrated that the quality of the robot model that is less important in presence of normal friction, is, instead, fundamental when the friction is significantly reduced. This evidence is relevant in defining the proper control strategy that can be limited to either linear model solutions (i.e., LMPC) or non-linear ones (i.e., NNMPC).  Completely frictionless conditions were not considered, since in such cases it would be impossible to move and steer unless adopting completely different locomotion solutions, for example sliding patterns towards specific directions, but this would deserve a different approach. 

Reviewer #2: if there are frictionless ground conditions, can't the underfoot be designed in such a way to improve contact?

Authors: As a follow-up on the previous response, the foot design is essential, for example the presence of bio-inspired solutions including claws on the feet is a possible solution to be investigated. As outlined before, this solution will increase the complexity of the mechanical structure and would be accompanied by a specific control strategies for the additional leg segments.

Reviewer 3 Report

This work presents an approach for locomotion control in a simulated Mini-Cheetah quadruped robot specifically for the task of moving in slippery terrain. The control strategy is designed as a hierarchical control structure. The low-level phase is achieved with CPG for general maneuvers including steering, and the high-level trajectory control using linear and nonlinear (neural network based) MPCs are compared. The research is overall solid and interesting with proper theoretical modeling and simulation validation. In my view, it should be sufficient for acceptance after the minor comments mentioned below.

·      The background discussion of MPC control should be updated with more up-to-date works in the field. MPC control is a very popular research field nowadays in both research and industry fields for complicated system control solutions, while most of the references are at least from more than 5 years ago, which have already been mentioned in the authors previous work published two years ago in

Arena, P., Patanè, L., Sueri, P. and Taffara, S., 2021. A data-driven neural network model predictive steering controller for a bio-inspired quadruped robot. IFAC-PapersOnLine54(17), pp.93-98.

·      The MPC design in section 2.2 seems to be similar to the design in the authors’ previous work mentioned above, if this work is based on an existing work, the authors should cite it properly and provide discussion of the differences between the two.

·      As the authors mentioned in 2.3.2, the existence of angular oscillation will introduce disturbance in the control process. Can the authors add discussion on how this issue is resolved?

·      Not sure if I missed it somewhere, in Fig. 7, how is the target output determined?

Author Response

Authors’ Reply to Reviewer 3’s comments

Reviewer #3: This work presents an approach for locomotion control in a simulated Mini-Cheetah quadruped robot specifically for the task of moving in slippery terrain. The control strategy is designed as a hierarchical control structure. The low-level phase is achieved with CPG for general manoeuvres including steering, and the high-level trajectory control using linear and nonlinear (neural network based) MPCs are compared. The research is overall solid and interesting with proper theoretical modelling and simulation validation. In my view, it should be sufficient for acceptance after the minor comments mentioned below.

 Authors: We thank the Referee for the time spent in reviewing our manuscript. We performed a careful revision of the paper to improve readability, better explaining the adopted techniques and the obtained results. Detailed answers to each aroused point are reported in the following.

Reviewer #3: ·      The background discussion of MPC control should be updated with more up-to-date works in the field. MPC control is a very popular research field nowadays in both research and industry fields for complicated system control solutions, while most of the references are at least from more than 5 years ago, which have already been mentioned in the authors previous work published two years ago in

Arena, P., Patanè, L., Sueri, P. and Taffara, S., 2021. A data-driven neural network model predictive steering controller for a bio-inspired quadruped robot. IFAC-PapersOnLine54(17), pp.93-98.

Authors: The authors’ intention was to report some dated works to testify that the MPC strategy is an effective control method well established in the literature. Following the Reviewer’s suggestion, other mostly recent references (2021-2023) were introduced to demonstrate the interest of the scientific community on the treated topics and to report the last results obtained.

As suggested by the Reviewer we duly indicated previous works from the authors on similar topics although the referred manuscript is substantially different from the here submitted paper. It contains a comparison among an LMPC and an NNMPC, both the focus and the case of study are completely different. In details, the paper referred deals with quadrupedal steering control in a quadruped robot completely different, both in structure and in motion controller, which there was based on a different CPG architecture.  Moreover, no discussion on the slippery conditions was done, which is the main focus of the submitted paper. Following the reviewer suggestion, the authors included this discussion on the revised manuscript.

Reviewer #3: ·      The MPC design in section 2.2 seems to be similar to the design in the authors’ previous work mentioned above, if this work is based on an existing work, the authors should cite it properly and provide discussion of the differences between the two.

Authors: Following the previous answer, the author included a discussion about the previous work outlining the relevant differences.

Reviewer #3: ·      As the authors mentioned in 2.3.2, the existence of angular oscillation will introduce disturbance in the control process. Can the authors add discussion on how this issue is resolved?

Authors: The author intention behind that sentence was to outline the intrinsic complexity in controlling  legged structure. In fact, the typical cyclic actuation of the legs and the alternation of stance and swing phases, unavoidably, especially during the touch down event, introduce periodic pulses which, filtered by the dynamics structure, lead to oscillations in the motion of the robot centre of mass. So, even while moving in perfectly flat terrains, the presence of angular oscillations due to this endogenous dynamic is present. As far as the robot speed is very low, this problem can be solved by acting on the robot balance. But typically, this is considered as a natural effect which is also important to take into account and maintain below a certain threshold. On other words, it is enough to control the distance between the vertical projection of the robot CoM on the ground and the boundary of the support polygon. 

Reviewer #3: ·      Not sure if I missed it somewhere, in Fig. 7, how is the target output determined?

Authors: Fig.7 report the effect of the model identification task. Given a specific steering command, through an angular velocity reference, the blue signal represents the real output shown by the robot, whereas the red and green traces report the linear and the NN response, respectively. It is noticed that the effect of the best linear model identified practically filters out all the robot oscillations. This fact obviously discards as noise an essential part of the dynamics, which is instead retained by the NN. This will be fundamental on the success of the predictive controller action.